# Feature integration within discrete time windows

Leila Drissi-Daoudi [1,2]*, Adrien Doerig[1,2] & Michael H. Herzog[1]

Sensory information must be integrated over time to perceive, for example, motion and melodies. Here, to study temporal integration, we used the sequential metacontrast paradigm in which two expanding streams of lines are presented. When a line in one stream is offset observers perceive all other lines to be offset too, even though they are straight. When more lines are offset the offsets integrate mandatorily, i.e., observers cannot report the individual offsets. We show that mandatory integration lasts for up to 450 ms, depending on the observer. Importantly, integration occurs only when offsets are presented within a discrete window of time. Even stimuli that are in close spatio-temporal proximity do not integrate if they are in different windows. A window of integration starts with stimulus onset and integration in the next window has similar characteristics. We present a two-stage computational model based on discrete time windows that captures these effects.

[1] Laboratory of Psychophysics, Brain Mind Institute, École Polytechnique Fédérale de Lausanne (EPFL), EPFL SV BMI LPSY, Station 19 CH-1015, Lausanne, Switzerland. [2] These authors contributed equally: Leila Drissi-Daoudi, Adrien Doerig. *email: leila.drissidaoudi@epfl.ch

A car runs through the night. The streetlights produce reflexions on its surface and its trajectory is partially occluded by trees and other cars. In addition, the information arriving at each retinal location is short and noisy. Hence, it is hard to estimate, for example, the car's color from single photoreceptor activity. An efficient way to estimate color is to average photoreceptor activities along the car's trajectory.

The brain indeed integrates information along motion trajectories as evident in the Sequential Metacontrast paradigm[1,2] (SQM). In the SQM, a central line is followed by pairs of flanking lines presented one after the other further and further away from the center (Fig. 1a). A percept of two moving streams diverging from the center is elicited (Supplementary Movie 1). The central line is invisible because it is masked by the subsequent lines[3,4]. Surprisingly, if the central line is offset, i.e., the lower segment is offset either to the right or left compared to the upper segment (this is called a vernier), the offset is visible at the subsequent stream even though the flanking lines themselves are aligned (Fig. 1, vernier condition). Observers attend to one of the streams and report the perceived offset direction. When, in addition to the central line, a flanking line is offset with an offset in the opposite direction, the offsets integrate and cancel each other (Fig. 1, vernier–anti-vernier condition). If both offsets are in the same direction (Fig. 1, vernier–pro-vernier condition), they add up and offset discrimination improves. Hence, features are non-retinotopically integrated across space and time.

Here, using the SQM, we show that spatio-temporal feature integration lasts up to 450 ms and is mandatory, i.e., observers are unable to report the offsets separately. Moreover, our data suggests that integration is not simply determined by spatiotemporal proximity, but rather occurs only when offsets are presented within a discrete window of time.

## Results

**Feature integration is mandatory and long lasting.** First, we show that feature integration is mandatory and long lasting (Experiment 1). The SQM was presented with 18 flanking pairs of lines (total stimulus duration: 750 ms). Performance is quantified in terms of dominance, i.e., the percentage of observers' responses in accordance with the central vernier offset (Fig. 2a). Before the experiment proper, and in all following experiments, we used an adaptive procedure[5] (PEST, see methods) to determine the individual offset sizes that led to a dominance of about 75% (condition V; Fig. 2a, blue diamonds) or 25% (condition AV; Fig. 2a, red diamonds). Next, sequences with two offsets, either in the same or opposite directions (conditions V-PV and V-AV, respectively), were presented. There was always a central vernier and a flank vernier at variable positions. In the first part of the experiment, participants were naive, i.e., they were not told that only a subset of lines in the display was offset nor how many lines were offset. Observers were instructed to attend to the left stream and report the perceived offset direction.

Dominance in the conditions V-PV was equal to or higher than in the condition with only the central offset (condition V; Fig. 2a, solid gray line). In conditions V-AV, the offsets canceled each other and dominance was around 50% except when the flank vernier was presented in frame 14, for which dominance was around 25% (Fig. 2a, solid black line). These results indicate that integration occurred up to 450 ms (frame 11). At 570 ms (frame 14), there was no integration anymore. In this case, observers reported the direction of the flank vernier.

In the second part of the experiment, the same observers were informed about the paradigm and instructed to ignore the flank offset and to report the central offset direction only (labeled [R1] for "Report 1st vernier"). At 290 ms (frame 7), all observers were not able to report the direction of the central vernier in the condition V-AV (Fig. 2a, dashed black line; two-sided paired t-tests: V-AV7 vs. V-AV7[R1]: $t(9) = 0.47$, $p = 0.65$, Cohen's

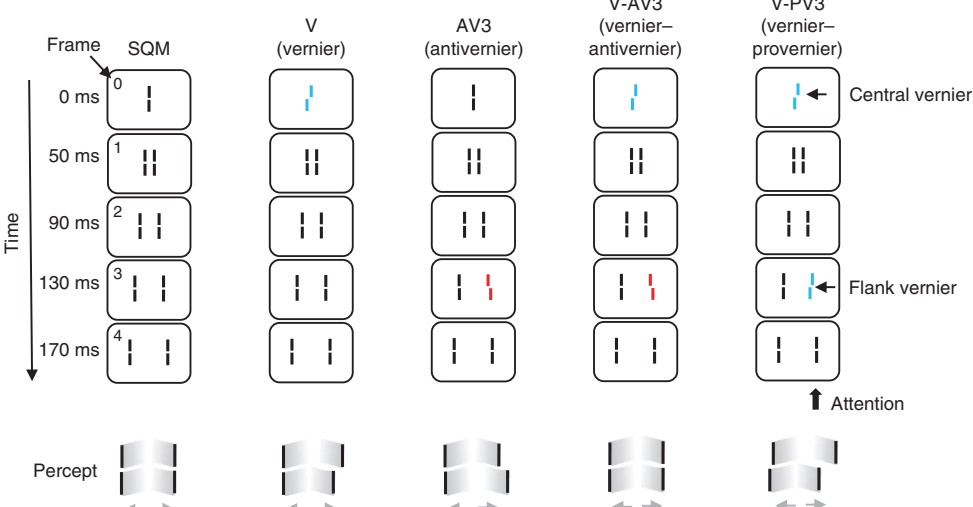

**Fig. 1** The Sequential Metacontrast paradigm (SQM). A central line is followed by pairs of flanking lines. Each line is presented for 20 ms, the inter-stimulus interval (ISI) is 20 ms (except the first ISI, which is 30 ms). A percept of two diverging streams is elicited. Observers attend to one of the streams (here, the right stream) and report the perceived offset direction (right/left) by pressing hand-held push-buttons. Condition V (vernier): only the central line is offset. The offset is visible at the following lines and observers report the offset direction. Condition AV (anti-vernier): only a flanking line is offset. The offset is visible in the attended stream. Condition V-AV (vernier-anti-vernier): the central line and one of the flanking lines are offset. The two offsets are in opposite directions and cancel each other. Observers cannot report the individual vernier offsets. Condition V-PV (vernier–pro-vernier): the central line and one of the flanking lines are offset. The two offsets are in the same direction and add up. Notation: For example, V-AV3 indicates that the central line and the flanking line in frame 3 are offset in opposite directions. The red and blue offset colors are for illustration purposes only. All elements were the same color (see Methods)

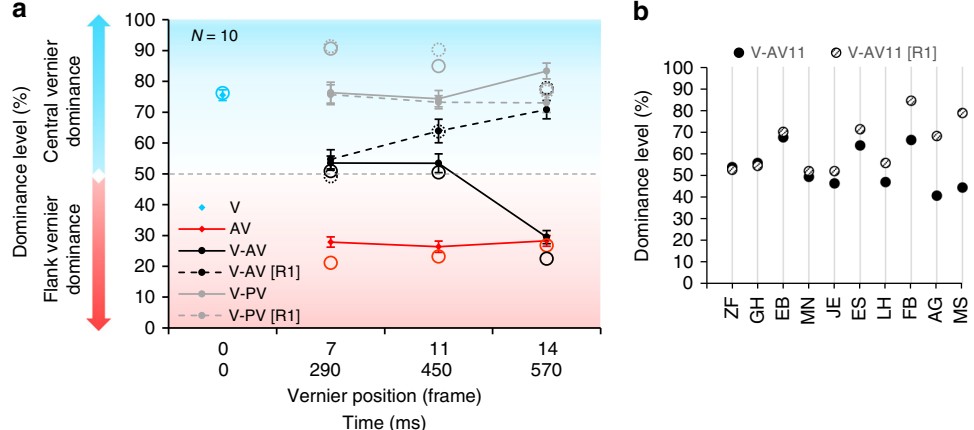

**Fig. 2** Results of experiment 1. **a** We presented the central vernier and, in addition, one flank vernier in frame 7, 11, or 14, respectively (290 ms, 450 ms, or 570 ms). Data is displayed as vernier dominance: the percentage of observers' responses in accordance with the central vernier. A dominance level above 50% (blue part of the plot) indicates that the central vernier dominates performance; a dominance level below 50% (red part of the plot) indicates that the anti-vernier dominates performance. In the first part of the experiment, observers were naive (solid lines). In the second part of the experiment, they were informed about the paradigm and instructed to report the central vernier offset ([R1], dashed lines). Mandatory integration lasts up to 450 ms, depending on the observer. Performance of a two-stage model (see Fig. 5) is presented by empty circles. The experimental data is well predicted. Error bars represent s.e.m. **b** Performance when the anti-vernier was presented at 450 ms (frame 11) for each observer in the naive condition V-AV11 (filled disks) and the informed [R1] condition V-AV11 [R1] (hashed disks). About half of the observers were able to report the direction of the central offset only (V-AV11 [R1]; observers ES-MS), whereas integration was mandatory for the other participants (observers ZF-JE). Thus, different observers have different integration window durations. Source data are provided as a Source Data file

$d = 0.15$, 95% CI [−0.77, 0.48]). Hence, integration was mandatory, i.e., observers could not access the individual offsets, even when explicitly trying. For some observers, mandatory integration lasted up to 450 ms, whereas for others mandatory integration was shorter, but at least 290 ms (Fig. 2b).

In additional experiments with 12 new observers, the flank offset was presented earlier, at frame 1 (50 ms), 2 (90 ms), 3 (130 ms), 5 (210 ms) or 7 (290 ms). Integration was mandatory for all observers in all of these conditions (Supplementary Fig. 1). In a further control experiment, we presented a visual cue either before or after each trial, which indicated which offset to report (central offset or flank offset). Results are similar as in experiment 1 (Supplementary Fig. 2).

**Features integrate only within the same discrete window.** Next, we show that vernier offsets integrate only when they are in the same discrete window of integration (Experiment 2). First, we presented the conditions V-AV8 (a central vernier and an anti-vernier in frame 8, i.e., at 330 ms) and V-AV12 (AV at 490 ms). Observers were informed about the paradigm and instructed to report the flank vernier (labeled [R2] for "Report 2nd vernier"). In condition V-AV8, the offsets integrated mandatorily (dominance of 52.2%, SEM = 3.0; Fig. 3b) confirming that both verniers were in the same window of integration. In condition V-AV12, observers were able to report the direction of the flank vernier (dominance of 27.4%, SEM = 3; Fig. 3b) showing that the flank vernier did not integrate with the central vernier. Next, we presented three offsets: a central vernier, an anti-vernier in frame 8, and a pro-vernier in frame 12 (condition V-AV8-PV12, Fig. 3a). Hence, two offsets were presented before 450 ms (central line and frame 8) and one offset after 450 ms (frame 12). Observers were instructed to report either the central vernier (condition [R1]) or the flank vernier (condition [R2]) but were not told that three lines were offset. When instructed to report the central vernier, dominance was close to 50%, indicating that the central vernier and the flank vernier in frame 8 (330 ms) integrated (there seems to be a small spill over from the flank vernier in frame 12 in condition V-AV8-PV12 [R1]; Fig. 3b). Hence, observers were not

able to report the central vernier separately. However, the flank vernier in frame 12 did not integrate with the other offsets. Observers were able to report its offset direction (condition V-AV8-PV12 [R2]; Fig. 3b). These results suggest that the flank vernier in frame 8 integrated with the central vernier but not with the vernier in frame 12, even though the flank offsets in frame 8 and 12 are closer in space and time (separated by 13.3′ and 160 ms) than the flank offset in frame 8 and the central offset (separated by 26.7′ and 330 ms). We suggest that the central vernier and the flank vernier in frame 8 integrate because they are in the same window of integration, but the flank vernier in frame 12 is in the next window and thus does not integrate with the previous offsets. Hence, integration occurs only within discrete windows. Even offsets that are in close spatio-temporal proximity do not integrate if they are in different windows.

**The "first" window of integration starts with stimulus onset.** Next, we show that the "first" window of integration starts with the presentation of the central line (Experiment 3). The central line was not offset. Otherwise, the flank offset positions were the same as in experiment 2 (frames 8 & 12). If integration starts with stimulus onset, the offsets in frames 8 and 12 should not integrate. If integration starts with the first task-relevant feature (i.e., the first offset), the offsets should cancel each other. The offsets were either in the same direction (PV8-PV12) or in opposite directions (PV8-AV12). Both conditions were presented randomly in the same block. Observers were instructed to either report the first offset ([R1]) or the second one ([R2]). Observers were able to report both offsets even when they were in opposite directions (Fig. 3c). These results indicate that the two offsets are in different integration windows and, hence, the first integration window starts with stimulus onset.

As a control, the same observers performed the same experiment but with the two offsets in the first window of integration (offsets in frame 1 and 5). Observers were not able to report the individual offsets, indicating mandatory integration of the two offsets (Supplementary Fig. 3). In this experiment, participants performed two training blocks with auditory error

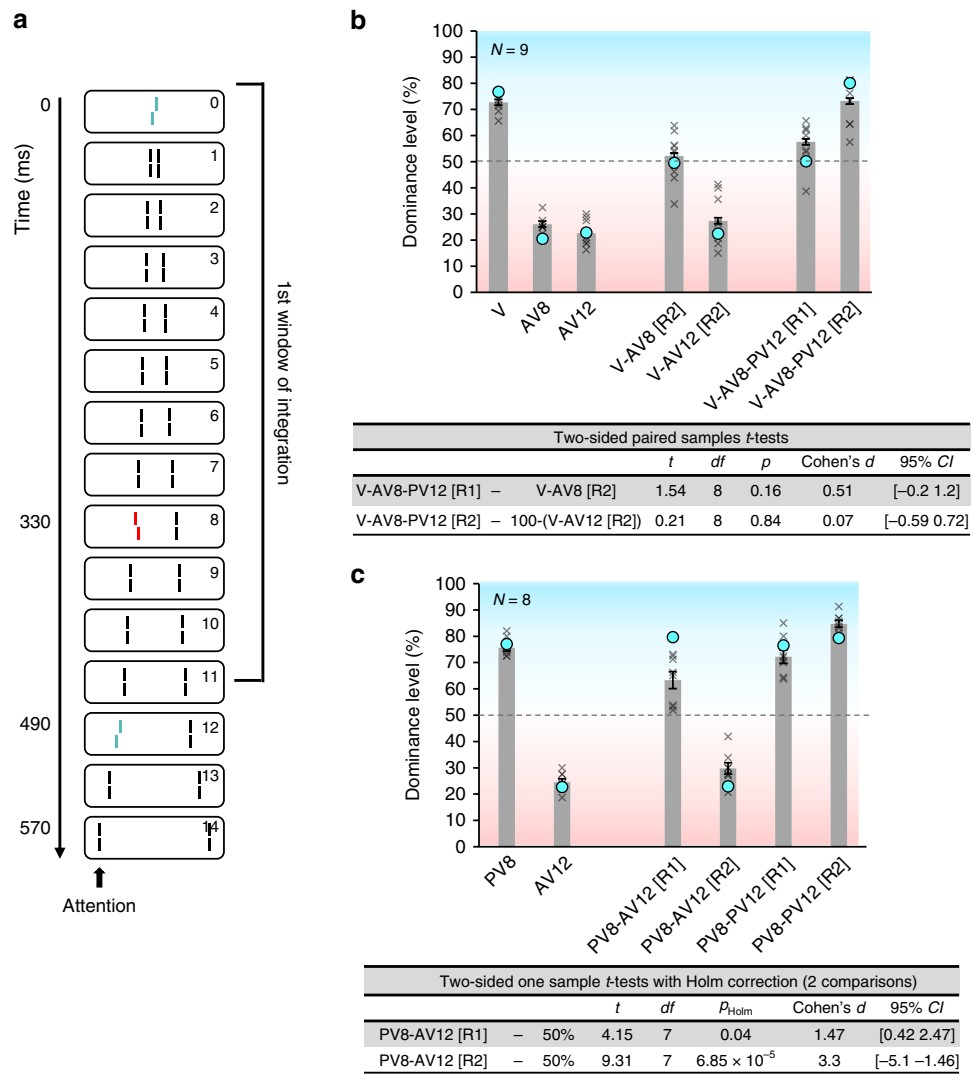

**Fig. 3** Experiments 2 and 3. **a** Experiment 2. A central vernier and an anti-vernier in frame 8 (330 ms) were presented before 450 ms. A pro-vernier was presented in frame 12 (490 ms), after 450 ms. **b** Results of experiment 2. In condition V-AV8 [R2], observers were not able to report the direction of the flank offset, suggesting mandatory integration. In condition V-AV12 [R2], observers were able to report the direction of the flank offset, suggesting that the flank offset did not integrate with the central vernier offset. We compare dominances in conditions V-AV8-PV12 [R1] and V-AV8-PV12 [R2] to V-AV8 [R2] and 100 − (V-AV12 [R2]), respectively, to test whether the addition of the third offset changed the integration. Observers were not able to report the direction of the central vernier in condition V-AV8-PV12 [R1], whereas they could report the direction of the pro-vernier in frame 12 (V-AV8-PV12 [R2]). We suggest that integration only occurs within discrete windows of integration. Even offsets that are in close spatio-temporal proximity do not integrate if they are in different windows. These results were well replicated by the model (see Fig. 5). Crosses indicate individual data. **c** Results of experiment 3. The flank verniers were in the same frames as in experiment 2, but there was no central vernier. When the flank verniers in frame 8 and 12 were in opposite directions, observers were able to report the individual offsets (PV8-AV12 [R1] and PV8-AV12 [R2]). Thus, the first window of integration seems to start with stimulus onset. These results are well replicated by the model (blue circles). Error bars represent s.e.m. Source data are provided as a Source Data file

feedback to ease understanding the task (the feedback was not provided during the experiment proper). In an additional experiment with 8 new observers, we removed these two training blocks with feedback. The results are similar (Supplementary Fig. 4).

**Integration in the subsequent window follows similar rules.** Lastly, we show that integration in the subsequent window follows similar rules of integration (Experiment 4). We extended the SQM to 20 pairs of flanking lines (830 ms stimulus duration). First, the streams were diverging from the center, then, from frame 10 on, they switched direction, converging back to the center (Fig. 4a). Five different conditions were presented. For

each condition, observers were instructed to give two responses at the end of each trial: first, the direction of the perceived offset at the beginning of the stream ([R1]), then the direction of the perceived offset at the end of the stream ([R2]; Fig. 4b). In condition 1, 3 lines were offset before 450 ms. Dominance level was close to 50%, showing that all offsets integrated. In condition 2, 3 lines were offset after 450 ms and all integrated (dominance level was close to 50%). In condition 3, three lines were offset before 450 ms, and an additional single line was offset after 450 ms. Only the offsets presented before 450 ms integrated, but the fourth offset was well reported, i.e., significantly above 50%. In condition 4, a single line was offset before 450 ms, and 3 lines were offset after 450 ms. The first offset could well be reported whereas all 3 offsets after 450 ms integrated. Lastly, in condition 5, three offsets

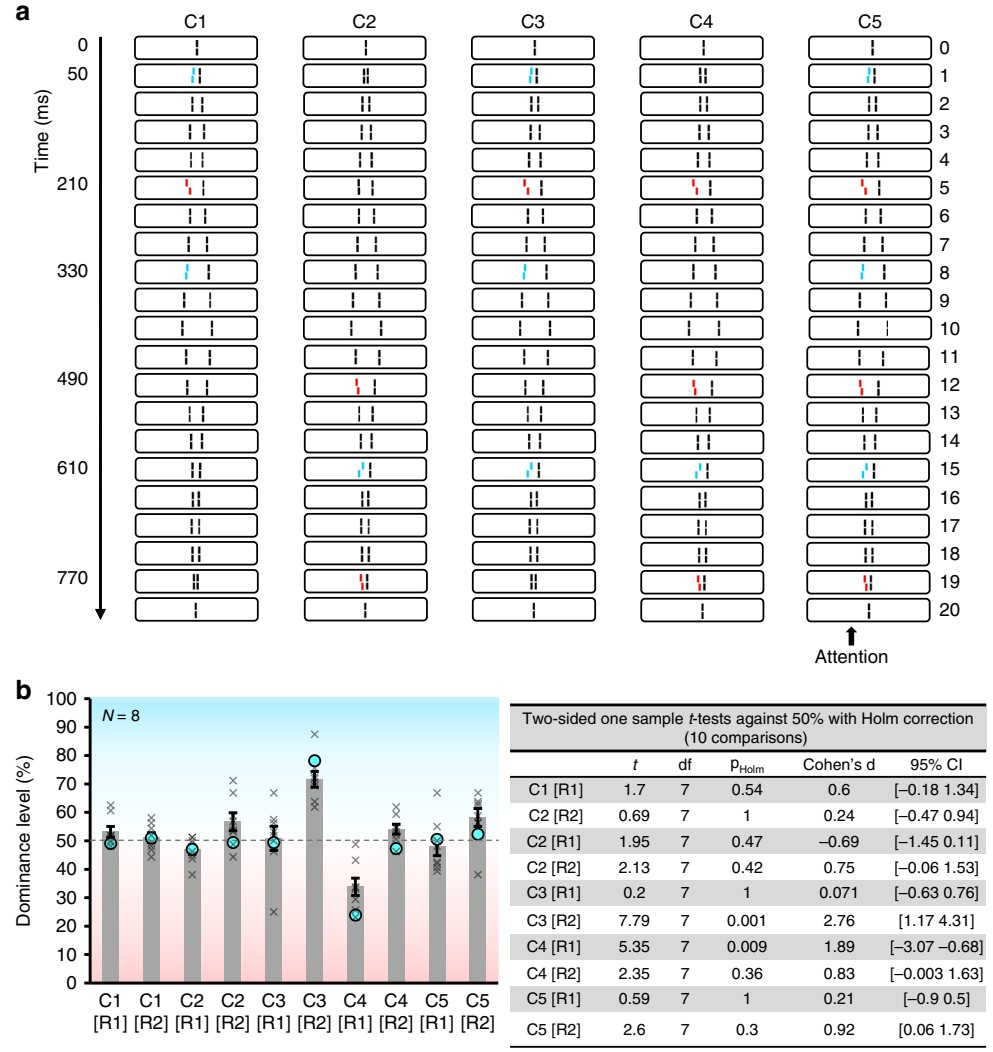

**Fig. 4** Experiment 4. **a** In the diverging part of the sequence (frames 0 to 10), the two pro-verniers (blue offsets) together have the same dominance as the anti-vernier (red offset) alone (see methods). In the converging part of the sequence (frames 10 to 20), the two anti-verniers (red offsets) have together the same dominance as the pro-vernier (blue offset) alone. **b** Verniers presented together either before or after 450 ms integrated (C1, C2, C3[R1], C4[R2] and C5). Only verniers that were presented alone either before or after 450 ms can be reported individually (C3[R2] and C4[R1]). Model outputs (see Fig. 5) are represented by the blue circles. Crosses indicate individual data. Error bars represent s.e.m. Source data are provided as a Source Data file

were presented before 450 ms and three offsets after. Dominance was close to 50% in both the [R1] and the [R2] conditions, indicating that the 3 verniers before 450 ms integrated and the 3 verniers after 450 ms integrated.

We also conducted the same experiment on 8 new observers with the streams originating from the center and diverging until the end of the stimulus, as in the classic SQM. The results are similar (Supplementary Fig. 5).

Together, these results indicate that (a) integration within the second time window is similar than in the first time window, and (b) verniers within a window always integrate, but not across windows.

## Discussion

Features in the SQM integrate mandatorily within a time window extending up to 450 ms (experiment 1), i.e., observers cannot separately report the offsets within this window. The duration of this window depends on the observer, but always extends over hundreds of milliseconds (Fig. 2b). Our results suggest that features that fall in different windows do not integrate, even if they are in close spatio-temporal proximity (experiment 2). The "first"

window of integration starts with stimulus onset (experiments 3). How and when a window starts under naturalistic conditions remains to be investigated. Once the first window "closes", a second similar window of integration opens (experiment 4).

These results suggest that perception is discrete. Features are mandatorily integrated within a window but there is little crosstalk between windows. It is not the spatiotemporal proximity *per se* that determines which elements integrate, but the "belongingness" to the same window. We previously proposed that features of objects, such as their form, color and duration, are continuously and unconsciously processed with high spatiotemporal resolution[6] (see also ref. [7] and refs. [8,9,10] for further discussion). At the end of the integration window, all features are perceived at once. Perception occurs only at discrete moments of time, sometimes even hundreds of milliseconds after stimulus onset. We can only speculate why windows last so long. We suggest that the brain needs to integrate information across space and time to detect changes, motion, etc. and solve the ill posed problems of vision. Several hundreds of milliseconds may be an ecologically optimal timescale for information integration.

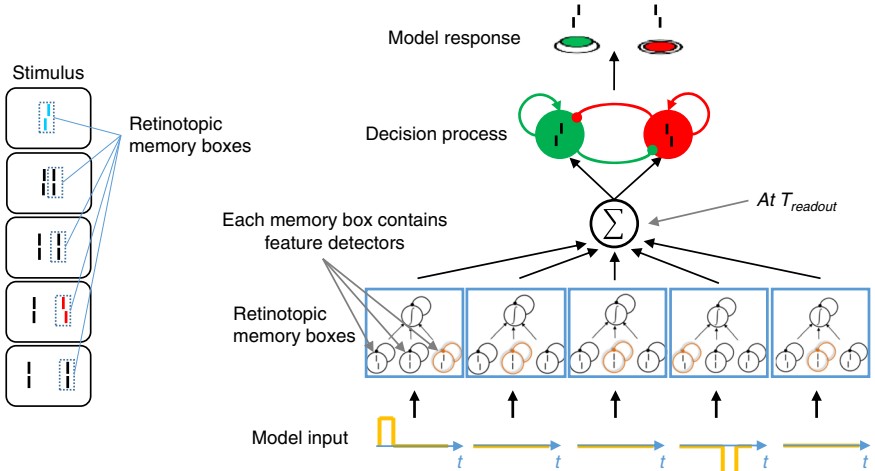

**Fig. 5** Computational model. Left: At each retinal location there is a memory box, which is activated when a visual feature is presented at this location. Right: When a visual feature appears at a given location, the memory box opens and processes information about the corresponding visual feature, i.e., a vernier with either a right, a left, or an aligned offset. These feature detectors are modeled as leaky integrators. We represent pro-verniers as $+1$, anti-verniers as $-1$, and aligned lines as 0. Once stimulation at this retinal location terminates, the memory box closes, buffering the integrated information[21]. Thus, information about visual features at each location is preserved throughout the discrete integration window. This processing is "unconscious". In this example, there are five memory boxes, and the input to each of them is shown at the bottom. At the end of a discrete time window (denoted $T_{readout}$), the content of the different memory boxes is combined, yielding the output of stage 1. In the present case, the attended stream of elements is perceived as a single moving object, so the outputs of all memory boxes are summed. Stage 2 receives the outputs of stage 1 and drives the decision. The task is to report vernier offset directions, which we implement using a biologically plausible decision making network proposed by Wong & Wang[22]. Details of the discrete computational model are provided in the methods

A long lasting period of integration is well in line with previous findings showing postdictive effects for several hundreds of milliseconds. For example, it has been shown that integration in RSVP tasks lasts up to 240 ms[11], that volition can disambiguate the percept of ambiguous stimuli up to 300 ms[12], that a cue, presented up to 400 ms after a visual stimulus, significantly increases the observer's capacity to detect and discriminate that stimulus[13], and that trans-magnetic stimulation modulates responses up to 420 ms in feature fusion[14].

It was shown that the temporal extent of feature integration can be modulated endogenously, by observers' expectations and attentional state[11,15]. Future work will investigate whether and how the window of integration in the SQM can be modulated by these and other endogenous factors and to what extent our findings generalize to other paradigms.

Importantly, features integrate in the SQM and do not mask each other because, first, observers can report the offsets when only a single vernier offset is present in the stream (conditions V and AV). Hence, the preceding and following straight lines do not render the vernier offsets invisible by masking. Second, dominance drops when the second vernier is offset in the opposite direction (conditions V-AV) and increases (above 75%) when the offsets are in the same direction (conditions V-PV).

Moreover, integration follows complex non-retinotopic rules and cannot be explained by low level aspects such as visual persistence, which usually lasts for <100 ms[16,17]. Instead, flexible grouping of elements is crucial. For example, integration occurs only within one stream and does not spill over to the other stream[1]. Furthermore, changing the grouping by removing lines can change integration strongly: the motion trajectory appears as interrupted, and there is no integration[2]. These findings are well in line with the Object Updating[17] account, which proposes that grouping is important to understand which features integrate. Hence, there are two complementary aspects: first, which visual elements are grouped together and therefore are prone to be integrated? Second, what are the temporal characteristics of this integration? We focused on the second question. In the SQM

paradigm, all elements of a stream are grouped and therefore integration is mandatory within a window. This gives us the possibility to study integration over long timescales, and therefore to study the temporal dynamics of integration.

To provide a proof of principle of our theory, we implemented a computational two-stage decision model, which integrates information within but not across windows (Fig. 5). Traditionally, drift diffusion models[18] are used to model decision making. In these models, evidence is integrated over time driving a decision variable (the evidence) toward one of two decision boundaries, corresponding to the two response alternatives. When a boundary is reached, a decision is made. Such one-stage models cannot explain our results because the evidence reaches the boundary for the first offset even before the second offset is presented. For this reason, in our two-stage model, first, information is unconsciously integrated and buffered during an extended period of time[19–21]. Second, at the end of the integration window, the percept occurs. More precisely, our model includes feature detectors at each retinal location. When a visual feature appears at a given location, a memory box opens and processes information about the corresponding visual feature in a leaky integrator[21]. Once stimulation at this retinal location terminates, the memory box closes, buffering the integrated information. Thus, information about visual features at each location is preserved throughout the discrete integration window. At the end of a discrete time window, the content of the different memory boxes is combined yielding the output of stage 1. Memory boxes pertaining to the same object are combined. In the present case, the attended stream of elements is perceived as a single moving object, so the outputs of all memory boxes are summed. Stage 2 receives the output of stage 1, and drives the decision of what will be perceived. We implemented the decision process using a biologically plausible network proposed by Wong and Wang[22]. This model successfully explained all our experimental findings (Fig. 2a empty circles, Figs. 3b, c and 4b blue circles).

To conclude, we showed that visual elements integrate only when they are in the same discrete temporal window. We propose

that perception occurs at discrete moments of time and presented a two-stage computational model based on discrete time windows to explain long-lasting feature integration. A discrete theory of conscious perception offers a straightforward explanation for postdictive effects[6], such as the flash-lag effect[23] and visual masking[3]. Our results are a first stepping-stone to understand the temporal structure of perception.

## Methods

**Observers**. Observers were students from EPFL and the University of Lausanne. Participants provided informed consent and had normal or corrected-to-normal vision. Visual acuity was tested with the Freiburg visual acuity test[24]. Observers were paid for their participation. The experiments were undertaken with the permission of the local ethics committee (Commission éthique du Canton du Vaud, protocol number: 164/14, title: Aspects fondamentaux de la reconnaissance des objets protocole général) and in accordance with the Declaration of Helsinki.

Ten observers took part in experiment 1 (age 22–30 years; 8 females). In experiment 2, 11 new observers participated (age 20–26; 4 females). Two participants were excluded from the analysis because of their dominance in conditions V-AV8 and V-AV12. These conditions served to make sure that the offsets in the central vernier and in frame 8 indeed integrated, and that the offsets in the central vernier and in frame 12 did not. Without this prerequisite, there was no sense in testing the V-AV8-PV12 [R1] and V-AV8-PV12 [R2] conditions. For these two observers, the dominance in condition V-AV8 indicated no integration (28.4%, SEM = 0.23). Thus, performance in condition V-AV12 should also have been in favor of the flank vernier, indicating no integration (because AV12 is further away from V than AV8). However, the performance was 45% (SEM = 3.5). Therefore, data was analyzed for 9 observers. Nine observers took part in experiment 3 (age 20–28; 3 females). One observer was excluded (8 were considered in the analysis) because her performance in conditions V-PV (two offsets in the same direction) was at chance, indicating random responses. In experiment 4, 8 observers (age 18–23; 4 females) participated.

**Apparatus**. Stimuli of experiments 1, 2, and 3 appeared on a HP-1332A XY-display equipped with a P11 phosphor controlled by a PC via a custom-made 16-bit DA interface. Line elements were composed of dots drawn with a dot pitch of 200 µm at a dot rate of 1 MHz. Dot pitch was chosen to make the dots slightly overlap so that the dot size was of the same magnitude as the dot pitch. Refresh rate was 200 Hz. Stimulus luminance was 80 cd/m² as measured with a Minolta LS-100 luminance meter by means of a two-dimensional dot grid using the aforementioned dot pitch and refresh rate. The room was dimly illuminated (~0.5 lux) and background luminance on the screen was below 0.5 cd/m². Viewing distance was 2 m and was kept constant by means of a chinrest.

In experiment 4, because of the spatial extent of the stimuli, they were presented on a BenQ XL2540 LCD monitor (1920 × 1080 pixels, 240 Hz; BenQ, Taipei, Taiwan) using MATLAB (R2013b, 64 bit, The MathWorks Inc., Natick, Massachusetts, United States) with Psychtoolbox[25,26]. Stimuli were white with a luminance of 98 cd/m², on a black background with a luminance of 0.1 cd/m². Participants were seated in a dimly lit room at 2.50 m from the screen.

**Stimuli**. The stimuli were variations of the sequential metacontrast stimulus[1,2] (SQM; for an animation see Supplementary Movie 1). The sequence started with a central line consisting of two vertical segments of a length of 10′ (arcmin) in experiments 1, 2, and 3 and separated by a vertical gap of 1′ (segment length of 20′ and vertical gap of 2′ in experiment 4). The line was followed by pairs of flanking lines presented one after the other further away from the center. The distance between the central line and the first flanking lines as well as between consecutive flanking lines was 3.3′. In experiments 1 and 2, the length of the first pair of flanking lines was 11.7′ and increased progressively by 1.7′ for the following lines. In experiment 3, the length of every flanking line was 20′. Each line was presented for 20 ms. The inter-stimulus interval (ISI) between the central line and the first pair of flanking lines was 30 ms and the ISI between consecutive pairs of flanking lines was 20 ms. A motion percept of two streams of lines diverging from the center is elicited.

One or more lines were spatially offset (vernier); that is, the lower segment of the line was offset to the right or to the left with respect to the upper segment. In experiments 1, 2, and 3, each trial was preceded by four markers at the corners and a fixation cross in the center of the screen for 500 ms followed by a blank screen for 200 ms. In experiment 4, each trial was preceded by a fixation dot in the center of the screen for 1 s followed by a blank screen for 500 ms. Then, the stimulus sequence was presented and participants responded by pressing one of two buttons.

**Offset calibration**. Before experiments 1, 2, and 3, the offset sizes were determined for each participant at each position in the stream to achieve comparable performance levels across observers. A PEST adaptive procedure[5] was used to determine offset sizes that yield around 75% performance (values are reported in the results' figures).

In experiment 4, offset sizes in frame 1 and 8 were simultaneously calibrated. Presented together with the same direction, these offsets yield 78% (SEM = 1.3) performance. The same was done for offsets in frame 12 and 19 (77%, SEM = 1.83). Offset sizes in frame 5 and 15 were determined individually to each yield around 75% performance when presented alone (73.7%, SEM = 1.6 and 78.1%, SEM = 1.54, respectively).

**Procedure**. The order of the blocks was randomized across observers to reduce the influence of hysteresis, learning or fatigue effects in the averaged data. For each observer, each condition was measured twice. After each block had been measured once, the order of the blocks was reversed for the second set of measurements. Each block contained 80 trials except for experiment 1, in which a block consisted of 100 trials (80 trials of condition V-AV and 20 trials of condition V-PV).

Depending on the experiment, observers were instructed to report the first and/or the second presented vernier offset ([R1] and/or [R2]). To explain the task to the participants, they were showed a printout of the stimuli and explained in detail what will appear on the screen. We made sure that they understood the paradigm and the task well.

Before experiment 3, participants performed two blocks containing an auditory error feedback. The blocks contained 40 V-AV trials and 40 V-PV trials. For one block, observers were instructed to report the first presented vernier offset ([R1]). For the other block, observers were instructed to report the second presented vernier offset ([R2]). We used this feedback, to help the observers with the task. The feedback was not present during the actual experiment. We conducted a control experiment (Supplementary Fig. 3), with the same procedure and with the same observers, but with the two offsets in the first window of integration (offsets in frames 1 and 5). Observers were not able to report the individual offsets, indicating mandatory integration of the offsets of frames 1 and 5. As an additional control, we also replicated the results of experiment 3 with 8 new observers (Supplementary Fig. 4) without the feedback.

In experiments 2–4, observers were instructed to report the first or the second offset blockwise. To rule out effects of this block design (e.g., expectations), we ran an experiment in which the participants were instructed either blockwise to report the first or the second offset, or a visual cue presented before each trial indicated the offset to report, or the cue appeared at the end of the trial. The pattern of results is similar in the three conditions (Supplementary Fig. 2).

**Data analysis**. Mean dominance level (see Fig. 2a) and standard error of the mean (SEM) across observers are computed for each condition.

We consider that a statistical test is significant when the p-value is below 0.05 after correction for multiple comparisons (Holm correction). As estimates of effect size, we report Cohen's d and 95% confidence interval, calculated with JASP software[27] (version 0.9.0.1).

**Power analysis**. The sequential metacontrast paradigm has been introduced in Otto et al.[1] in which the effect size Cohen's d ≈ 2.0. To achieve a power of 90%, a sample size of 5 observers is needed. To be "safe", between 8 and 10 observers participated per experiment. The smallest effect size of a significant result in our experiments is 1.47 (experiment 3). With a sample size of 8 observers, we achieved a power of 94.3%. The power analysis was computed with the G*Power software[28] (version 3.19.2).

**Model**. The model was implemented in MATLAB. Stimuli are modeled by a time-varying input signal stim(t), which is +1 during the presentation of pro-verniers, −1 for anti-verniers, and 0 otherwise. During stage 1, feature integration occurs within memory boxes. Each visual element of the SQM is processed in its own memory box (see main text, Fig. 4). The input to each memory box is subjected to leaky integration: $\frac{dE}{dt} = \frac{-E}{\tau} + stim$, yielding the integrated evidence E. $\tau$ is the integration time constant. Since stimuli have high contrast, the evidence integration is modeled as a noise-free process. Each memory box closes and buffers its current integrated evidence value when the element that opened it disappears.

Stage 2 starts at $T_{readout}$, which is the integration window duration. $T_{readout}$ is the model's second parameter. The integrated evidence is summed across all memory boxes. The input to the decision network is drawn from a normal distribution centered around the summed evidence, multiplied by a scalar gain: decision network input = $N(c * \sum_i memorybox_i(T_{readout}), \sigma)$, where c is the gain, and $\sigma$ is the noise distribution's standard deviation. For the decision network itself, we used the neural network by Wong & Wang[22], based on code provided by the authors that we modified to fit in our model pipeline. This model is a simplified two-variable version of a biophysically plausible decision neural network. Pro-vernier and anti-vernier are each represented by a neural population and a decision is made when one of these populations reaches an activity threshold. This network has only one free parameter $\mu_0$, which determines the input to the network in the absence of stimuli. All other parameters are based on physiological data. All implementation details and equations are given in the accompanying code.

The parameters $\tau$, c, $\sigma$ and $\mu_0$ were fitted by hand using the results of experiment 1. The same parameter values were used for all simulations ($\tau = 0.3$, $c = 0.3$, $\sigma = 0.2$ and $\mu_0 = 0.2$). The window length $T_{readout}$ was set to 425 ms for all simulations. The only exception was condition [R1] of experiment 1, where different observers had

different window durations: in this case, we assigned a window length of 475 ms to observers ZF-JE to account for their prolonged integration windows (see Fig. 2b; all other parameters were kept the same). The code is available at https://github.com/adriendoerig/Feature-integration-within-discrete-time-windows-code.

**Reporting Summary**. Further information on research design is available in the Nature Research Reporting Summary linked to this article.

## Data availability
The data are available from the corresponding author upon reasonable request. The source data underlying Figs. 2a, b, 3b, c and 4b and Supplementary Figs. 1a, b, 2a, b, 3b, 4 and 5b are provided as a Source Data file.

## Code availability
The code of the computational model is available at https://github.com/adriendoerig/Feature-integration-within-discrete-time-windows-code.

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

## Acknowledgements
We are grateful to Simona Garobbio for collecting the data of experiment 3 and fitting the parameters of the computational model, to Marc Repnow for technical support, and to Kong-Fatt Wong and Xiao-Jing Wang for giving us their code of the decision process used in our model. This work was supported by a grant from the Swiss SystemsX.ch initiative (2015/336) and by the Swiss National Science Foundation grant 'Basics of visual processing: from elements to figures' (176153).

## Author contributions
L.D.-D., A.D., and M.H.H. designed the study. L.D.-D. collected and analyzed the data. A.D. coded the computational model. L.D.-D., A.D., and M.H.H. wrote the manuscript.

## Competing interests
The authors declare no competing interests.
