## [Peer Review File · Nature Communications]

Reviewers' Comments:

Reviewer #1:

Remarks to the Author:

This paper presents a highly interesting set of studies providing evidence for windows of integration that are discrete, suggesting that conscious perception might also be discrete, integrating across fixed windows of integration to form a given percept. This is demonstrated by using sequences of stimuli which include Vernier display, that affect the overall perception of the sequence.

I found the findings novel, and the entire paper intriguing and thought provoking. I would be glad to see it published in Nature Communications. I do have some comments though, which should be addressed by the authors prior to such publication (some of them cast doubt on the interpretation of the findings, and are accordingly critical). I am certain they would make the manuscript stronger and even more interesting.

Major comments

1. Subjects' task is not very clear to me, especially for the [R2] condition. Given the speed of the display and the similarity between the stimuli, how were subjects explained about what they need to do (which, here, was to report the 2nd Vernier)? How do they know which is the 2nd Vernier on which they have to respond?

2. More importantly, this task (reporting 1st/2nd Vernier, as opposed to the first task that relates to the overall perceived offset direction) raises the concern that chance performance might not require any integration, but simply involve masking. That is, subjects could be at chance not because they integrated the stimuli and the resulting percept involved a cancelling out of the first and second Verniers, but rather because the 2nd Vernier was rendered invisible due to feedforward and feedback masking from adjacent stimuli. And so, I am not sure how these results prove the claim the authors make (again, as opposed to the holistic task). Could the authors please clarify?

3. Sample sizes are very small (which is not necessarily a problem, given that this is a psychophysical experiment and that the variability seems to be very low). But how were these sample sizes determined? Can the authors at least provide an estimation of the power of these experiments? Also, the authors excluded 3 subjects whose behavior was 'incoherent' or 'at chance'. I am not sure this is warranted, as it seems as if the exclusion was not made based on an independent criterion but simple because their behavior did not comply with the effect itself. This is quite circular; can the authors justify this choice?

4. If I understand the procedure correctly, the experiment was blocked with each condition taking place in a different block (correct? If not, please make the text clearer). Why use a block design here? Doesn't this introduce effects of expectations etc.? If the claim is that integration is a mandatory part of perception, wouldn't it make more sense to mix the trials and see how perception is affected in a more bottom-up manner?

5. Why did Experiment 3 necessitate feedback? And how was a 'correct' response defined in this feedback? The concern here is that subjects actually learned to disintegrate the two Verniers, rather than do it naturally.

6. The authors keep referring to the 450ms window. But is this warranted? In line 83, "lasting up to 450ms" sounds as if this happens for all subjects, while it actually only occurs for some. Is this enough to make the claim that this is the size of the window? Also, could the authors speculate as to why the window lasts 450ms? Are there any biological constraints/mechanisms that could support this claim?

Minor comments

7. The opening of the abstract feels like an opening of the paper, which to me (and this is a matter of style), felt a bit out of place. More importantly, though, the connection between the car example and the next sentence which talks about the experiment was not very clear. Thus, I don't think that in its current phrasing, the car example facilitates the understanding of the paper and its

topic. But it is of course up for the authors to decide if they want to change it or not.

8. Figure 1 is very helpful for understanding the design, but the illustrations of the percept are not clear enough. That is, the differences between the percepts are not pronounced enough, in my opinion, to make the reader understand the effect on perception. I suggest making them a bit less subtle.

9. The authors aimed at 75% correct responses. But what is considered correct here? As the 'ground truth' is actually different than the perception, I am not sure the word 'correct' makes sense here and would replace it to avoid confusion.

10. Line 75: I found the use of words "Even at an SOA of 290ms" misleading, as it sounds as if this happens not only for 290ms but also for other SOAs, and this is not the case. I advise to rephrase.

Reviewer #2:

Remarks to the Author:

Review of Drissi-Daoudi et al. "Feature integration within discrete time windows"

Summary

The authors report a series of experiments in which feature integration in a sequential Vernier offset task is measured. In the critical conditions, observers are asked to detect the orientation offset of an initial central Vernier stimulus, which is followed after a variable interval by a competing flanking Vernier. If the Verniers are integrated over time (and space), they should cancel each other out, resulting in a lack of any Vernier perception on the attended side. The results showed that integration indeed occurs and is relatively long-lasting, has a discrete time-window, and is triggered (has its onset) at the appearance of the first stimulus. A computational model built by the authors is able to fit these outcomes.

Comments

The inferences made by the authors are interesting. Typically, temporal integration is not thought to last as long as 450 ms, and spatiotemporal proximity is thought to play an important role. Also the idea of a series of discrete integration time windows, initiated by stimulus onset is intriguing. I do have three critical notes about these inferences, however.

1) Temporal extent. With regard to the temporal window of integration, I am not fully convinced that it lasts as long as claimed by the authors. From Figure 1b it is apparent that naïve subjects (solid black line) do not exhibit central nor flanker Vernier dominance up to 450 ms, which would suggest they are integrating the stimuli. But after instruction (dashed black line), the subjects are able to discern the central Vernier; they start to show central dominance at 450 ms. In other words, they could either completely separate the Verniers from the beginning, perceptually, or somehow disentangle them post-hoc. Integration is only truly unavoidable ("mandatory") at 290 ms, where a lack of dominance is still observed for the informed subjects. The fact that the window is longer than as reported on average for some subjects (as shown in the supplementary material) does not change the fact that the main conclusion should be that temporal integration lasts for 290 ms on average, not 450 ms.

2) Spatial extent. In the current paradigm, integration takes places across different spatial locations. While this is interesting, it must be noted somewhere that the nature of the paradigm does at the very least facilitate this effect. The perception of consistent motion moving from a central location outward (or vice versa) produces correspondence between the locations, which would not be the case in sequential displays that do not generate such motion perception. In fact, it is conceivable that the spatial effect is carried entirely by this special property of the paradigm. Can the authors exclude this possibility?

3) Discrete windows. The authors demonstrate nicely that two events that occur close in time

(position 8 and position 12) may not integrate if the first of those is already part of an ongoing event (with the starting Vernier). Nevertheless, temporal proximity does undoubtedly play an important role in temporal integration. To state that "...what matters for integration is not spatiotemporal proximity." (page 5) is surely an overstatement. To wit, once enough time has passed, a second stimulus will no longer integrate with the first (which happens between the first Vernier and the second in frame 11-14; see Exp 1). It is more accurate to state only that integration occurs in, or is governed by, discrete windows, without reference to the temporal property.

There is some literature on temporal integration that should be addressed. First, the authors claim that their long window of integration (but see #1) is well in line with previous findings. They draw on some of their own work, as well as on studies of other phenomena (ambiguous stimuli, detection performance), not directly related to integration itself. However, there is also an extensive literature on integration in partial report and dot-array integration tasks, which measure temporal integration quite directly (e.g., Hogben & Di Lollo, 1974; Eriksen & Collins, 1967). This literature consistently shows far shorter periods of temporal integration, which must be reconciled somehow with the present findings. Second, we have observed that the onset as well as the length of the window of integration is under a degree of endogenous control (Akyürek, Toffanin, & Hommel, 2008; Akyürek & Wolff, 2016). In the RSVP tasks we used here, integration was found to last up to 240 ms (actually comparable to the present results, in my view), but would also depend on the expected speed of the stimuli in our task, and on the nature of the stimuli (target or non-target). We observed, for instance, that integration was completely halted when a non-target stimulus intervened. These studies should be weighed in the discussion, as the claims currently made by the authors are rather absolute with regard to the conditions and spatiotemporal extent of integration, while they may actually be more flexible than the current data suggest.

Minor

In Figure 1 the rightmost stream seems to show a black Vernier on the left side of the flank Vernier display, where there should be none, which might be confusing.

Because the number of observers is on the low side as well as variable, it may be helpful to state whether statistical power computed a priori, and if so, what N was minimally needed to detect the expected effects.

Signed,
Elkan Akyürek

Reviewer #3:

Remarks to the Author:

In this manuscript, the authors sought to examine how feature integration occurs, specifically, whether feature integration is determined by spatiotemporal proximity, or by discrete windows (defined by stimulus onsets/offsets).

To investigate this, the authors obtained behavioural data from human participants in a sequential metacontrast paradigm, and also implemented a computation model.

Concerns:

The authors interpret the results from their sequential metacontrast masking paradigm as reflecting the temporal window of feature-integration. Results such as these are often interpreted within a different framework – that of object-updating (e.g., Enns, J. T., Lleras, A., & Moore, C. M. (2010). Object updating: A force for perceptual continuity and scene stability in human vision. In

R. Nijhawan & B. Khurana (Eds.), *Space and Time in Perception and Action* (pp. 503-520). Cambridge: Cambridge University Press.). It is unclear and unsubstantiated that the updating of object representations truly operationalises feature integration.

In a similar vein, the authors make very general conclusions about how vision works one from the results of a single paradigm. This is not justified. The results may indicate more about sequential metacontrast masking than how vision operates more generally. It is not possible to say without converging evidence from other measures/approaches.

Small number of participants. This is especially problematic where they are drawing conclusions from null results. E.g. "observers were not able to report the direction of the central vernier in condition V-AV... $t(5) = 1.4$, $p = .23$, Cohen's $d = .56$ ". These stats lead one to think that with a larger sample size, this comparison may well have become significant.

Novelty is not well substantiated. The role of stimulus offsets in visibility in masking is already well-documented, see: Macknik, S. L., & Livingstone, M. S. (1998). Neuronal correlates of visibility and invisibility in the primate visual system. *Nature Neuroscience*, 1, 144-149. doi:10.1038/393

Reviewers' comments:

Reviewer #1 (Remarks to the Author):

This paper presents a highly interesting set of studies providing evidence for windows of integration

that are discrete, suggesting that conscious perception might also be discrete, integrating across fixed windows of integration to form a given percept. This is demonstrated by using sequences of stimuli which include Vernier display, that affect the overall perception of the sequence. I found the findings novel, and the entire paper intriguing and thought provoking. I would be glad to see it published in Nature Communications. I do have some comments though, which should be addressed by the authors prior to such publication (some of them cast doubt on the interpretation of the findings, and are accordingly critical). I am certain they would make the manuscript stronger and even more interesting.

Thank you very much for your insightful comments that, we agree, helped make the manuscript stronger.

Major comments

1. Subjects' task is not very clear to me, especially for the [R2] condition. Given the speed of the display and the similarity between the stimuli, how were subjects explained about what they need to do (which, here, was to report the 2nd Vernier)? How do they know which is the 2nd Vernier on which they have to respond?

Thank you for pointing out this lack of clarity. We showed the participants a printout of the stimuli and explained them in detail what will appear on the screen. We made sure that they understood the paradigm and the task well. Indeed, all participants reported that they understood our explanation clearly. Hence, we are confident that they knew and understood what they had to report.

This is also experimentally evident. When the 2nd offset is presented at 570ms, observers can report the individual offsets showing that they understood and performed the task correctly. Moreover, in experiment 3, the offsets are close to each other. Still, observers can report both offsets individually.

We clarified how the task was explained in the "Procedure" subsection of the Methods.

2. More importantly, this task (reporting 1st/2nd Vernier, as opposed to the first task that relates to the overall perceived offset direction) raises the concern that chance performance might not require any integration, but simply involve masking. That is, subjects could be at chance not because they integrated the stimuli and the resulting percept involved a cancelling out of the first and second verniers, but rather because the 2nd Vernier was rendered invisible due to feedforward and feedback masking from adjacent stimuli. And so, I am not sure how these results prove the claim the authors make (again, as opposed to the holistic task). Could the authors please clarify?

This is an important point. There are three reasons why offsets *integrate* and do not mask each other. First, observers can easily report the offsets when only a single vernier is present in the stream (e.g. condition V). Hence, the preceding and following *straight* lines do not render the verniers invisible by masking. Second, performance only drops when the second vernier is offset

in the *opposite direction* (conditions V-AV). Third, performance increases (> 75%) when the offsets are in the same direction (conditions V-PV), clearly indicating that offsets integrate.

3. Sample sizes are very small (which is not necessarily a problem, given that this is a psychophysical experiment and that the variability seems to be very low). But how were these sample sizes determined? Can the authors at least provide an estimation of the power of these experiments? Also, the authors excluded 3 subjects whose behavior was ‘incoherent’ or ‘at chance’. I am not sure this is warranted, as it seems as if the exclusion was not made based on an independent criterion but simple because their behavior did not comply with the effect itself. This is quite circular; can the authors justify this choice?

Sample size. Thank you for this comment. You are right, we ought to have provided our estimation of the power.

The sequential metacontrast paradigm has been introduced in Otto et al. (2006), in which the effect size had a Cohen’s $d \approx 2$, which is very large according to Cohen. To achieve a power of 90%, a sample size of 5 observers is needed. To be “safe”, we decided to invite between 6 and 9 participants per experiment. The smallest effect size of a significant result in our experiments is 1.47 (experiment 3). With a sample size of 6 observers, we have a power of 81.8%.

To be on the safe side, we added 4 observers to experiment 1 (which previously had 6 observers). Therefore, we now have a minimum of 8 observers in all experiments. The achieved power (with an effect size of 1.47) is 94.3%.

We now present the power analysis in the Methods section.

Exclusion of observers. We did not exclude observers based on conditions that are crucial for the results. We agree that this would be circular. Rather, we based exclusion only on conditions, which are *prerequisites* for the experiment proper. We performed these prerequisites to make sure that participants did perform the task correctly and to exclude observers when this was not the case.

In experiment 3, one observer was responding randomly as evident by the dominance level that was at 50% when both verniers were offset in the same direction. Since each offset individually was calibrated to yield 75% dominance, dominance should at *least* have been around 75%.

In experiment 2, the crucial conditions are V-AV8-PV12 [R1] and V-AV8-PV12 [R2] because we wanted to test whether the flank offset in frame 8 integrates with the central offset or with the flank offset in frame 12 when all 3 offsets are presented. For this comparison to make sense, it was crucial that in conditions V-AV8 [R2] and V-AV12 [R2] the central offset indeed integrated with the one in frame 8, and that the central offset did not integrate with the one in frame 12. Without this prerequisite, it makes no sense to test the V-AV8-PV12 [R1] and V-AV8-PV12 [R2] conditions. For two observers, the dominance in condition V-AV8 indicated no integration (28.4%, $SEM = 0.23$). If the reason was that the integration window was shorter for these observers (which can be possible), then the dominance in condition V-AV12 would also have to indicate no integration

because the flank offset is further away than in V-AV8. However, this was not the case. Dominance was 45% ($SEM = 3.5$). Hence, we excluded these two observers.

4. If I understand the procedure correctly, the experiment was blocked with each condition taking place in a different block (correct? If not, please make the text clearer). Why use a block design here? Doesn't this introduce effects of expectations etc.? If the claim is that integration is a mandatory part of perception, wouldn't it make more sense to mix the trials and see how perception is affected in a more bottom-up manner?

You are right, we used blocked conditions. We used this design because the task is already quite difficult. Even though we use a blocked design, expectation effects cannot influence our results because there were V-AV (offsets in opposite directions) and V-PV (offsets in the same direction) trials in each block. Hence, expectations should not matter since the results depend on the offsets of *all* the verniers in the display, which were randomized.

To fully rule out an effect of a block design, we conducted an additional experiment comparing blocked and interleaved trials, as you suggested. We used a central offset plus a flank offset either in frame 5 or 14. In the blocked condition, the participants were instructed to report either the first (central) or the second (flank) offset for all stimuli in a block. In the interleaved condition, a visual cue indicated which offset to report. We tested presenting the cue either before each trial or at the end of each trial. The pattern of results is similar in all conditions. Dominance is generally slightly lower in the interleaved condition, presumably because the task is harder since participants need to attend to both the cue and the SQM stimulus, but, clearly, the same conclusions hold. The results are provided in the supplementary material.

5. Why did Experiment 3 necessitate feedback? And how was a 'correct' response defined in this feedback? The concern here is that subjects actually learned to disintegrate the two Verniers, rather than do it naturally.

When we piloted the experiment (3 participants), we found that observers could report the second offset in frame 12 very well whereas reporting the offset in frame 8 was a bit more difficult. This shows again that the offsets did not integrate (otherwise observers would *not* have been able to report the second offset). We introduced two blocks with feedback to help observers with the task. In one of these blocks, observers were instructed to report the first offset, and in the other block they had to report the second offset. Auditory feedback was given when the response was not correct. For example, when observers had to report the first offset, if the offset was to the right and the observer pressed "left", the sound was played (the second offset was randomly offset either to the right or to the left). We controlled that the feedback did not have an effect (such as breaking integration) by having the exact same experiment but with both offsets in the first window of integration (a central offset and a flank offset in frame 5; same observers). In this case, observers were not able to report either offset showing that the feedback did not interfere with integration (experiment in supplementary materials).

To really rule out the role of feedback, we ran a new experiment without the two blocks with feedback with 8 new observers. The results are similar showing that actually the feedback was not needed. The results are now provided in the supplementary materials.

6. The authors keep referring to the 450ms window. But is this warranted? In line 83, “lasting up to 450ms” sounds as if this happens for all subjects, while it actually only occurs for some. Is this enough to make the claim that this is the size of the window? Also, could the authors speculate as to why the window lasts 450ms? Are there any biological constraints/mechanisms that could support this claim?

You are right, the window depends on observer. For some participants, integration *is still* mandatory at 450ms (i.e. dominance is the same whether they are naïve or not). For others, integration is mandatory at 290ms, but no longer at 450ms. This is why we said the window can last *up to* 450ms (depending on the participant). To clarify this point, we added a figure (Figure 2b) to show the individual data.

Regarding the reason why the window can last up to 450ms, we can only speculate. To our knowledge, there is no known neural mechanism that can straightforwardly be linked to our findings. On a more conceptual level, the brain needs to integrate information across space and time to detect changes, motion, etc. and solve the ill posed problems of vision. This requires prolonged integration times but this integration cannot go on forever. Several hundreds of milliseconds may be an ecologically optimal timescale for information integration.

We now discuss these points in the manuscript.

Minor comments

7. The opening of the abstract feels like an opening of the paper, which to me (and this is a matter of style), felt a bit out of place. More importantly, though, the connection between the car example and the next sentence which talks about the experiment was not very clear. Thus, I don't think that in its current phrasing, the car example facilitates the understanding of the paper and its topic. But it is of course up for the authors to decide if they want to change it or not.

Thank you. We completely rewrote the abstract and introduction.

8. Figure 1 is very helpful for understanding the design, but the illustrations of the percept are not clear enough. That is, the differences between the percepts are not pronounced enough, in my opinion, to make the reader understand the effect on perception. I suggest making them a bit less subtle.

Thank you, we modified the figure accordingly.

9. The authors aimed at 75% correct responses. But what is considered correct here? As the ‘ground truth’ is actually different than the perception, I am not sure the word ‘correct’ makes sense here and would replace it to avoid confusion.

You are right, it is confusing.

We changed 75% correct by 75% dominance.

10. Line 75: I found the use of words “Even at an SOA of 290ms” misleading, as it sounds as if this happens not only for 290ms but also for other SOAs, and this is not the case. I advise to rephrase.

Thank you, we rephrased it.

Moreover, in the supplementary material, we show the results of additional experiments that show that integration also happens with flank verniers at different SOAs between 50ms and 290ms (50ms, 90ms, 130ms, 210ms and 290ms SOA).

Reviewer #2 (Remarks to the Author):

Review of Drissi-Daoudi et al. “Feature integration within discrete time windows”

Summary

The authors report a series of experiments in which feature integration in a sequential Vernier offset task is measured. In the critical conditions, observers are asked to detect the orientation offset of an initial central Vernier stimulus, which is followed after a variable interval by a competing flanking Vernier. If the Verniers are integrated over time (and space), they should cancel each other out, resulting in a lack of any Vernier perception on the attended side. The results showed that integration indeed occurs and is relatively long-lasting, has a discrete time-window, and is triggered (has its onset) at the appearance of the first stimulus. A computational model built by the authors is able to fit these outcomes.

Comments

The inferences made by the authors are interesting. Typically, temporal integration is not thought to last as long as 450 ms, and spatiotemporal proximity is thought to play an important role. Also the idea of a series of discrete integration time windows, initiated by stimulus onset is intriguing. I do have three critical notes about these inferences, however.

Thank you for your insightful comments!

1) Temporal extent. With regard to the temporal window of integration, I am not fully convinced that it lasts as long as claimed by the authors. From Figure 1b it is apparent that naïve subjects (solid black line) do not exhibit central nor flanker Vernier dominance up to 450 ms, which would suggest they are integrating the stimuli. But after instruction (dashed black line), the subjects are able to discern the central Vernier; they start to show central dominance at 450 ms. In other words, they could either completely separate the Verniers from the beginning, perceptually, or somehow

disentangle them post-hoc. Integration is only truly unavoidable (“mandatory”) at 290 ms, where a lack of dominance is still observed for the informed subjects. The fact that the window is longer than as reported on average for some subjects (as shown in the supplementary material) does not change the fact that the main conclusion should be that temporal integration lasts for 290 ms on average, not 450 ms.

You are right. This was not clear in the previous version of the manuscript. For some participants, integration *is still* mandatory at 450ms (i.e., dominance is the same whether they are naïve or not). For others, integration is not mandatory any longer at 450ms but is mandatory at 290ms. Hence, their durations must be somewhere between 290ms and 450ms. We show the individual data in Figure 2b now.

2) Spatial extent. In the current paradigm, integration takes places across different spatial locations. While this is interesting, it must be noted somewhere that the nature of the paradigm does at the very least facilitate this effect. The perception of consistent motion moving from a central location outward (or vice versa) produces correspondence between the locations, which would not be the case in sequential displays that do not generate such motion perception. In fact, it is conceivable that the spatial effect is carried entirely by this special property of the paradigm. Can the authors exclude this possibility?

You are completely right. The properties of the paradigm allow the integration across different locations. For example, when removing stream elements, which leads to a non-smooth trajectory, the individual verniers do not integrate. It is the spatio-temporal smoothness of the trajectory that leads to the integration of the features. Accordingly, perceptual grouping plays a strong role in determining which elements are grouped in the SQM (Otto, Oğmen & Herzog, 2006).

We added these points to a new paragraph in the discussion.

3) Discrete windows. The authors demonstrate nicely that two events that occur close in time (position 8 and position 12) may not integrate if the first of those is already part of an ongoing event (with the starting Vernier). Nevertheless, temporal proximity does undoubtedly play an important role in temporal integration. To state that “...what matters for integration is not spatiotemporal proximity.” (page 5) is surely an overstatement. To wit, once enough time has passed, a second stimulus will no longer integrate with the first (which happens between the first Vernier and the second in frame 11-14; see Exp 1). It is more accurate to state only that integration occurs in, or is governed by, discrete windows, without reference to the temporal property.

Yes, you are right. We never intended this meaning. We rephrased the sentences to avoid this confusion. Here are the rephrased sentences:

“Even stimuli that are in close spatio-temporal proximity do not integrate if they are in different windows.”

“Moreover, our data suggests that integration is not simply determined by spatiotemporal proximity, but rather occurs only when offsets are presented within a *discrete* window of time.”

“Even offsets that are in close spatio-temporal proximity do not integrate if they are in different windows.”

“It is not the spatiotemporal proximity *per se* that determines which elements integrate, but the “belongness” to the same window”

There is some literature on temporal integration that should be addressed. First, the authors claim that their long window of integration (but see #1) is well in line with previous findings. They draw on some of their own work, as well as on studies of other phenomena (ambiguous stimuli, detection performance), not directly related to integration itself. However, there is also an extensive literature on integration in partial report and dot-array integration tasks, which measure temporal integration quite directly (e.g., Hogben & Di Lollo, 1974; Eriksen & Collins, 1967). This literature consistently shows far shorter periods of temporal integration, which must be reconciled somehow with the present findings.

Thank you for these literature suggestions. We think that integration in the above paradigms is mainly based on low-level retinotopic mechanisms, such as visual persistence, whereas in our paradigm integration is non-retinotopic. The SQM is not about the retinotopic superposition of sub-patterns into a larger pattern, but an integration of features along a motion trajectory. The single lines are not visible during the presentation but their offsets are “extracted” and stored. Thank you for pointing this out. We now discuss these studies and argue that visual persistence in the SQM does not play a crucial role because the above studies have shown rather short visual persistence.

Moreover, we discuss cases in which integration breaks down and link this to grouping in a new paragraph. In a nutshell, there are two complementary aspects: first, which visual elements are grouped together and therefore are prone to be integrated (the studies you mention study this process). Second, what are the temporal characteristics of this integration? We focused on the second question. In the SQM paradigm, all elements are grouped and therefore integration is mandatory and long lasting. This gives us the possibility to study integration over long timescales, and therefore to study the temporal dynamics of integration.

Second, we have observed that the onset as well as the length of the window of integration is under a degree of endogenous control (Akyürek, Toffanin, & Hommel, 2008; Akyürek & Wolff, 2016). In the RSVP tasks we used here, integration was found to last up to 240 ms (actually comparable to the present results, in my view), but would also depend on the expected speed of the stimuli in our task, and on the nature of the stimuli (target or non-target). We observed, for instance, that integration was completely halted when a non-target stimulus intervened. These studies should be weighed in

the discussion, as the claims currently made by the authors are rather absolute with regard to the conditions and spatiotemporal extent of integration, while they may actually be more flexible than the current data suggest.

Thank you for pointing out these interesting results, which we now cite (sorry that we were not aware of them). We state that integration may be under endogenous control.

Minor

In Figure 1 the rightmost stream seems to show a black Vernier on the left side of the flank Vernier display, where there should be none, which might be confusing.

Thank you for spotting this, we corrected it.

Because the number of observers is on the low side as well as variable, it may be helpful to state whether statistical power computed a priori, and if so, what N was minimally needed to detect the expected effects.

Thank you for this comment. You are right, we ought to have provided our estimation of the power.

The sequential metacontrast paradigm has been introduced in Otto et al. (2006), in which the effect size had a Cohen's $d \approx 2$, which is very large according to Cohen. To achieve a power of 90%, a sample size of 5 observers is needed. To be "safe", we decided to invite between 6 and 9 participants per experiment. The smallest effect size of a significant result in our experiments is 1.47 (experiment 3). With a sample size of 6 observers, we achieved a power of 81.8%.

To be on the safe side, we added 4 observers to experiment 1 (which previously had 6 observers). Therefore, we now have a minimum of 8 observers. The achieved power (with an effect size of 1.47) is 94.3%.

We now show our power analysis in the Methods section.

Signed,

Elkan Akyürek

Reviewer #3 (Remarks to the Author):

In this manuscript, the authors sought to examine how feature integration occurs, specifically, whether feature integration is determined by spatiotemporal proximity, or by discrete windows (defined by stimulus onsets/offsets).

To investigate this, the authors obtained behavioural data from human participants in a sequential metacontrast paradigm, and also implemented a computation model.

Concerns:

The authors interpret the results from their sequential metacontrast masking paradigm as reflecting the temporal window of feature-integration. Results such as these are often interpreted within a different framework – that of object-updating (e.g., Enns, J. T., Lleras, A., & Moore, C. M. (2010). Object updating: A force for perceptual continuity and scene stability in human vision. In R. Nijhawan & B. Khurana (Eds.), *Space and Time in Perception and Action* (pp. 503-520). Cambridge: Cambridge University Press.). It is unclear and unsubstantiated that the updating of object representations truly operationalises feature integration.

Thank you for this relevant citation, which we included in our manuscript.

Enns et al. indeed make some similar points to ours, notably about how grouping can be used for object-level integration. However, their main point is quite different from ours and addresses a very different question. They focus on what determines which elements integrate and propose that object-level cues are crucial. In our contribution, the main aim is to address a complementary question: when elements integrate, what are the spatiotemporal characteristics of this integration? For this reason, the question about whether or not object-updating truly operationalizes feature integration is orthogonal to our main question. We use a paradigm in which we know that features integrate, and study the spatiotemporal characteristics of this integration. We clarified this point in a new paragraph in the discussion.

In a similar vein, the authors make very general conclusions about how vision works one from the results of a single paradigm. This is not justified. The results may indicate more about sequential metacontrast masking than how vision operates more generally. It is not possible to say without converging evidence from other measures/approaches.

You are right. Our results speak only for the SQM, which is particularly well suited because the integration is mandatory and long lasting. In the discussion, we make a link with other studies using different paradigms that may support our conclusions. Of course, more work, and different paradigms, are needed to investigate to what extent our findings generalize. We added a clarification in the discussion.

Small number of participants. This is especially problematic where they are drawing conclusions from null results. E.g. “observers were not able to report the direction of the central vernier in condition V-AV... $t(5) = 1.4$, $p = .23$, Cohen’s $d = .56$ ”. These stats lead one to think that with a larger sample size, this comparison may well have become significant.

Thank you for this comment. You are right, we ought to have provided our estimation of the power.

The sequential metacontrast paradigm has been introduced in Otto et al. (2006), in which the effect size had a Cohen's $d \approx 2$, which is very large according to Cohen. To achieve a power of 90%, a sample size of 5 observers is needed. To be "safe", we decided to invite between 6 and 9 participants per experiment. The smallest effect size of a significant result in our experiments is 1.47 (experiment 3). With a sample size of 6 observers, we achieved a power of 81.8%.

To be on the safe side, we added 4 observers to experiment 1 (which previously had 6 observers). Therefore, we now have a minimum of 8 observers. The achieved power (with an effect size of 1.47) is 94.3%.

We now show our power analysis in the Methods section.

Novelty is not well substantiated. The role of stimulus offsets in visibility in masking is already well-documented, see: Macknik, S. L., & Livingstone, M. S. (1998). Neuronal correlates of visibility and invisibility in the primate visual system. *Nature Neuroscience*, 1, 144-149. doi:10.1038/393

Our research question is quite different from the one of Macknik and Livingstone. Macknik and Livingstone studied how elements are rendered invisible by back- and forward masking and determined the underlying neural correlates. However, this is not the question of our manuscript where we study how multiple (invisible) elements integrate and how long mandatory integration lasts. To the best of our knowledge, investigating discrete time windows of perception using well controlled psychophysical experiments has not been done.

We now cite Macknik and Livingstone (1998) in the introduction when introducing backward masking.

Reviewers' Comments:

Reviewer #1:

Remarks to the Author:

The authors addressed all my concerns in a satisfactory manner. I commend the authors for going the extra mile and running more experiments to examine potential problems. I find this paper very interesting, and would love to see published in Nature Communications.

Reviewer #2:

Remarks to the Author:

Review of Drissi-Daoudi et al. "Feature integration within discrete time windows" (NCOMMS-19-14555A).

Comments

This is a revision of a manuscript I reviewed previously. I had raised some issues with the original manuscript. The first main point was the temporal extent of the integration presently observed. The authors have clarified this point, acknowledging that for some observers the length of integration may be shorter than for others. Second was the possible contribution of the spatial nature of the design. The authors have now clarified this and made it explicit. Third was the issue of discrete windows and the importance of temporal proximity. The authors have rephrased this appropriately. Finally, I pointed at some literature that I thought to be relevant. The authors have now incorporated these suggestions.

In summary, the authors have addressed all of my points in a satisfactory manner. I am thus supportive of the current version.

Reviewer #3:

Remarks to the Author:

I thank the authors for taking the time to respond to my comments. However, I still have one outstanding concern. Perhaps I was too indirect in bringing up the possibility of object-updating (I don't think that they convincingly refute this point, but anyway), but I'll cut to the chase: the authors claim in their response that "We use a paradigm in which we know that features integrate." Please justify this. What is the independent evidence that this paradigm truly measures feature integration?

Dear Editor,

Thank you very much for the opportunity to revise our manuscript and address the “new” concern of reviewer 3. Below we respond to this comment. We now address this issue in the discussion (highlighted).

Reviewer #3 (Remarks to the Author):

I thank the authors for taking the time to respond to my comments. However, I still have one outstanding concern. Perhaps I was too indirect in bringing up the possibility of object-updating (I don't think that they convincingly refute this point, but anyway), but I'll cut to the chase: the authors claim in their response that "We use a paradigm in which we know that features integrate." Please justify this. What is the independent evidence that this paradigm truly measures feature integration?

Thank you for your comment. This is indeed an important point that was also previously raised by reviewer #1. There are three reasons why offsets integrate and do not mask each other in the SQM paradigm. First, observers can report the offsets when only a single vernier offset is present in the stream (conditions V and AV). Hence, the preceding and following straight lines do not render the vernier offsets invisible by masking. Second, dominance only drops when the second vernier is offset in the opposite direction (conditions V-AV). Third, dominance increases (above 75%) when the offsets are in the same direction (conditions V-PV). This clearly indicates that offsets integrate.

We now include these arguments in the manuscript and hope that we have addressed your concern in a satisfactory manner.

Reviewers' Comments:

Reviewer #3:

Remarks to the Author:

I am satisfied with the authors' response.